# The Physical Behaviour Intensity Spectrum and Body Mass Index in School-Aged Youth: A Compositional Analysis of Pooled Individual Participant Data

**DOI:** 10.3390/ijerph19148778

**Published:** 2022-07-19

**Authors:** Stuart J. Fairclough, Liezel Hurter, Dorothea Dumuid, Ales Gába, Alex V. Rowlands, Borja del Pozo Cruz, Ashley Cox, Matteo Crotti, Lawrence Foweather, Lee E. F. Graves, Owen Jones, Deborah A. McCann, Robert J. Noonan, Michael B. Owen, James R. Rudd, Sarah L. Taylor, Richard Tyler, Lynne M. Boddy

**Affiliations:** 1Movement Behaviours, Nutrition, Health & Wellbeing Research Group, Department Sport & Physical Activity, Edge Hill University, St Helens Road, Ormskirk L39 4QP, UK; coas@edgehill.ac.uk (A.C.); tylerr@edgehill.ac.uk (R.T.); 2Research Institute of Sport and Exercise Science, Liverpool John Moores University, Liverpool L3 3AF, UK; l.hurter@ljmu.ac.uk (L.H.); crotti.mc@gmail.com (M.C.); l.foweather@ljmu.ac.uk (L.F.); l.e.graves@ljmu.ac.uk (L.E.F.G.); o.r.jones1@2017.ljmu.ac.uk (O.J.); d.a.mccann@2014.ljmu.ac.uk (D.A.M.); s.l.taylor1@ljmu.ac.uk (S.L.T.); l.m.boddy@ljmu.ac.uk (L.M.B.); 3Alliance for Research in Exercise, Nutrition and Activity, Allied Health and Human Performance, University of South Australia, Adelaide, SA 5001, Australia; dot.dumuid@unisa.edu.au; 4Centre for Adolescent Health, Murdoch Children’s Research Institute, Melbourne, VIC 3052, Australia; 5Faculty of Physical Culture, Palacký University Olomouc, CZ 779 00 Olomouc, Czech Republic; ales.gaba@upol.cz; 6Diabetes Research Centre, Leicester General Hospital, University of Leicester, Gwendolen Road, Leicester LE5 4PW, UK; alex.rowlands@leicester.ac.uk; 7National Institute for Health Research (NIHR) Leicester Biomedical Research Centre (BRC), University Hospitals of Leicester NHS Trust and the University of Leicester, Leicester LE5 4PW, UK; 8Department of Sports Science and Clinical Biomechanics, University of Southern Denmark, 5230 Odense, Denmark; bdelpozocruz@health.sdu.dk; 9Department of Psychology, University of Liverpool, Liverpool L69 7ZA, UK; r.noonan@liverpool.ac.uk; 10Department of Applied Health and Social Care and Social Work, Faculty of Health, Social Care and Medicine, Edge Hill University, Ormskirk L39 4QP, UK; michael.owen@edgehill.ac.uk; 11Department of Teacher Education and Outdoor Studies, Norwegian School of Sport Sciences, 0863 Oslo, Norway; jamesr@nih.no

**Keywords:** accelerometer, CoDa, children, adolescents, physical activity, adiposity, intensity spectrum

## Abstract

We examined the compositional associations between the intensity spectrum derived from incremental acceleration intensity bands and the body mass index (BMI) z-score in youth, and investigated the estimated differences in BMI z-score following time reallocations between intensity bands. School-aged youth from 63 schools wore wrist accelerometers, and data of 1453 participants (57.5% girls) were analysed. Nine acceleration intensity bands (range: 0–50 mg to ≥700 mg) were used to generate time-use compositions. Multivariate regression assessed the associations between intensity band compositions and BMI z-scores. Compositional isotemporal substitution estimated the differences in BMI z-score following time reallocations between intensity bands. The ≥700 mg intensity bandwas strongly and inversely associated with BMI z-score (*p* < 0.001). The estimated differences in BMI z-score when 5 min were reallocated to and from the ≥700 mg band and reallocated equally among the remaining bands were −0.28 and 0.44, respectively (boys), and −0.39 and 1.06, respectively (girls). The time in the ≥700 mg intensity band was significantly associated with BMI z-score, irrespective of sex. When even modest durations of time in this band were reallocated, the asymmetrical estimated differences in BMI z-score were clinically meaningful. The findings highlight the utility of the full physical activity intensity spectrum over a priori-determined absolute intensity cut-point approaches.

## 1. Introduction

Overweight and obesity in children and adolescents (hereafter referred to as youth) continue to increase in prevalence [1] and are significant risk factors for health and wellbeing [2,3,4]. Consequently, there has long been a research focus on the relationships between youth physical behaviours and adiposity [5]. This research demonstrates a favourable relationship between physical activity and adiposity, with the most consistent associations observed for moderate-to-vigorous physical activity (MVPA) and vigorous physical activity (VPA) [6,7]. Conversely, unfavourable associations between adiposity and sedentary screen time (television viewing in particular) are commonly reported [8].

Accelerometers are used in health research to estimate the duration and mode of different physical behaviours [9]. The focus of epidemiological and intervention studies has typically been on the total volume or bouted time spent in specific physical behaviours and intensities. Traditional accelerometer data reduction methods rely heavily on cut-points whereby the same absolute intensity thresholds are applied to all participants’ data to generate estimates of time spent above these thresholds. These resultant free-living physical activity estimates are prone to intensity misclassification and bias because the cut-points used are specific to the original calibration protocols and sample populations [10]. In addition, the condensing of continuous data into only a small number of pre-specified intensity categories (e.g., sedentary time, light physical activity (LPA), MVPA) causes an important loss of information from the captured accelerometer data because a very limited portion of the data is used [11]. In recognition of these factors, emerging analytical approaches for assessing the associations between accelerometer-determined physical behaviours and health outcomes were recently advocated in the GRANADA consensus [9]. 

One such approach is to use a wide range of incremental acceleration intensity bands to describe daily physical behaviour patterns across the full intensity spectrum [9]. This gives high-resolution descriptions of the full physical activity intensity pattern and allows for an examination of the relationships between health outcomes and a wider range of intensities [11]. The small number of intensity spectrum studies conducted to date in school-aged youth have observed higher intensity activity to be most beneficially associated with health outcomes [11,12,13,14]. The inverse relationship between physical activity intensity and duration dictates that levels of higher intensity physical activity are typically low (e.g., 0.6% to 3.9% of waking hours [15,16]). Arguably, this is one reason why higher intensity physical activity has been less emphasised in physical activity promotion messaging, which focuses more on physical activity of at least a moderate intensity [17], which is considered more accessible and attainable for population health. However, even small amounts of higher intensity physical activity are beneficial for cardiorespiratory fitness in youth [18], and it has been shown in adults that similarly low doses may be favourably associated with other outcomes related to cardiometabolic health [19], such as adiposity [7]. Moreover, a range of structured and unstructured opportunities exist for youth to accumulate time in higher intensity physical activity, including sports participation, physical education classes, and time-efficient exercise modalities such as high intensity interval training (HIIT) [18]. 

Compositional data analysis with isotemporal substitution accommodates the complexity of analysing the intensity spectrum and allows the theoretical effects of displacing fixed durations of time between mutually exclusive incremental intensity bands to be investigated [9]. Almost all of the previous accelerometer-derived compositional analysis studies in school-aged youth have used three- or four-part compositions as the physical behaviour exposure and have relied on the traditional cut-points approach [20,21,22,23,24,25,26,27]. These studies described waking-hours physical behaviours as sedentary time, LPA, and MVPA and applied various accelerometer metrics and data reduction methods to produce time-use estimates of activity. An ActiGraph count cut-point for an MVPA of 2296 counts·min^−1^ was applied to describe the compositional associations between physical behaviours and adiposity across 24 h [25] and the school day [20]. Other studies used a much lower MVPA count cut-point of 1499 counts·min^−1^ [24,27], while raw acceleration cut-points (e.g., 200 mg; [21]) have also been applied to describe the compositional associations between physical behaviours and adiposity. Each of these studies reported significant associations between activity compositions and the adiposity outcomes, which included BMI z-score, percent and absolute body fat, and waist circumference. Moreover, a common finding in each study was that the greatest predicted differences in adiposity were when the time was reallocated to and from MVPA relative to the other intensities.

In contrast, only one dataset, using a sample of 10-year-old children with ActiGraph counts data, has examined the full intensity spectrum in school-aged youth [11,14]. Focusing on 5-to-16-year-olds, our novel study extends this work by employing compositional data analysis across the physical activity intensity spectrum, using raw acceleration data obtained from different accelerometer brands. The aims of this study were to (i) examine the compositional associations between the intensity spectrum derived from multiple incremental raw acceleration intensity bands and body mass index (BMI) z-score in youth and (ii) investigate the estimated differences in BMI z-score when durations of time were reallocated between incremental intensity spectrum bands.

## 2. Materials and Methods

### 2.1. Data Acquisition and Study Eligibility

Ethically approved wrist accelerometry studies led or supervised by the first or last authors were identified for inclusion within this pooled individual participant data analysis. Eligible studies involved school-aged youth who participated in observational or interventional physical activity research studies during school term time. Inclusion in the analysis studies required non-intervention assessments of wrist accelerometer-derived physical behaviours; thus, baseline data were used for contributing intervention studies. In addition, studies needed to provide anthropometric and demographic data including age, sex, and area-level socioeconomic status. Where published, details of these studies can be found elsewhere [22,28,29,30,31,32,33]. Investigators with a major involvement in the eligible studies (e.g., past PhD students, co-supervisors) were approached by email and invited to contribute individual participant data to allow for data harmonisation and subsequent pooled analysis. On receipt of the signed data transfer agreements, all the contributing investigators transferred their de-identified data via a secure file sharing system. Ethical approval for this pooled analysis study was granted by Edge Hill University’s Science Research Ethics Committee (#ETH2021-0034). The data were available from ten studies conducted in 63 schools between 2015 and 2019 in the Merseyside, Lancashire, and Greater Manchester regions of northwest England.

### 2.2. Outcomes

#### 2.2.1. Outcome Variable and Covariates

BMI was calculated from stature and body mass measured using standard procedures, with participants wearing light clothing and no shoes [34]. BMI z-scores were assigned using British 1990 growth reference data [35], and International Obesity Task Force BMI cut-points were applied to classify participants by weight status [36]. Socioeconomic status (SES) was measured at the neighbourhood level using the English Indices of Multiple Deprivation (EIMD) [37,38] based on home postcodes. EIMD deciles were generated, where smaller values indicated a lower SES. Group (school) mean centering was applied to the participants’ ages to aid the model interpretation and to reduce the risk of multicollinearity.

#### 2.2.2. Physical Behaviour Acceleration Exposure Variables

In the contributing studies, ActiGraph GT9X (ActiGraph, Pensacola, FL, USA; seven studies), GENEActiv Original (Activinsights, Cambs, UK; two studies), and Axivity AX3 (Axivity Ltd., Newcastle-Upon-Tyne, UK; one study) triaxial accelerometers were used. The devices have a dynamic range of ±8 g and were requested to be worn for up to seven consecutive days on the non-dominant wrist using either 24 h (seven studies) or waking hours wear protocols (two studies), with the sampling frequency set at 100 Hz (seven studies) or 30 Hz (two studies). The devices were initialised and data were downloaded using the latest releases of the respective ActiLife (versions 6.13.1 to 6.13.4), GENEActiv (versions 2.2 to 3.1), and OMGUI software (version 43) available at the time of data collection. Physical behaviour metrics were generated from the raw accelerometer data files (ActiGraph: gt3x followed by conversion to .csv format; GENEActiv: .bin format; Axivity: .wav format) and were processed in R using the package GGIR version 1.11-0 [39]. 

The data were harmonised by re-processing the raw accelerometer files using a standardised waking hours ‘day’ of 07:00 to 23:00 h (16 h or 960 min) to enable the inclusion of those studies that did not use a 24 h wear protocol. Signal processing included autocalibration, using local gravity as a reference [40], the detection of implausible values, and the detection of non-wear. Non-wear was imputed by default in GGIR, whereby invalid data were imputed by the average at similar time points on other days of the week [41]. The participants’ accelerometer data were excluded from the analyses if the post-calibration error was >10 milligravitational units (mg) and/or if <3 days of valid wear (i.e., ≥600 min·day^−1^) were available. We calculated the average magnitude of dynamic acceleration (i.e., average acceleration) as the Euclidean norm of the three accelerometer axes, with 1 *g* subtracted and negative values truncated to zero (ENMO) [42] averaged over 1-s epochs and expressed in mg. The average acceleration from the three devices worn on the non-dominant wrist has demonstrated equivalence in adults [43]. We generated nine acceleration bands using 50 mg increments for the first seven bands and wider bands for the remaining two (0–50 mg, 50–100 mg, 100–150 mg, 150–200 mg, 200–250 mg, 250–300 mg, 300–350 mg, 350–700 mg, and ≥700 mg). Based on the empirical evidence, the acceleration bands that are 350–700 mg and ≥700 mg reflect the upper range of MVPA [44] and higher intensity activities such as jogging and running [45], respectively. They were combined to reflect the anticipated very short duration (or absence) of accumulated accelerations within 50 mg bands above 350 mg [10]. The average minutes per day spent in each intensity band were calculated to create a nine-part composition per participant. The time in the nine bands was summed to 960 min. 

### 2.3. Data Analysis

In these analyses, the outcome variable was the BMI z-score, and the exposure variables were the proportions of time spent in each of the nine acceleration intensity bands. Data were available for 1803 participants. Of these, 67 cases with missing BMI z-scores were removed because under a missing at a random assumption, there is no advantage in multiple imputation for missing data on the outcome variable [46]. Of the remaining 1736 participants, 233 did not achieve the accelerometer minimum wear criteria. These cases were also removed because the imputation of these summary activity estimates would rely on too many unknown assumptions about the pattern of missingness and would thus introduce random variation. No significant differences were observed between the included and excluded participants for centred age (*p* = 0.89), BMI z-score (*p* = 0.89), school type (*p* = 0.28), and sex (*p* = 0.15). A higher proportion of participants who did not achieve the accelerometer wear criteria were low SES (i.e., EIMD deciles 1–3, 66.5% vs. 55.1% who met the criteria), while fewer were high SES (EIMD deciles 7–10, 9.3% vs. 23.3%). Home postcodes were missing for 50 of the remaining 1503 participants (3.3% of the data), which prevented the calculation of EIMD deciles; these cases were also removed, leaving an analytical sample of N = 1453. 

Compositional analyses were conducted using the R package compositions (v. 1.40-5, Mathsoft, Cambridge, MA, USA) [47]. Nine-part time-use compositions were expressed as nine specific sets of eight isometric log-ratios (ILRs) [48], which were used in multivariate linear regression models. BMI z-score was the dependent variable, and the intensity band composition ILRs were the explanatory variables. Each set of ILRs contained one ILR_1_ (i.e., the first pivot coordinate), which captured the time in one specific intensity band relative to all the remaining bands (i.e., the geometric mean of the remaining intensity bands), ensuring that each of the nine intensity bands were considered against all the remaining bands. The models were adjusted for sex, centred age, SES, accelerometer model, and accelerometer recording frequency. The influence of school on the models was trivial (ICC = 0.03), and the model parameter VIF values ranged from 1.01 to 1.96, indicating no multicollinearity. Where sex was significantly associated with BMI z-score, follow-up sex-stratified analyses were performed to assess the moderating influence of sex. The sex-specific models included the same covariates as the full sample models, with sex removed. 

If the ANOVA table of the model fit showed that the set of intensity band ILRs was significantly associated with BMI z-score (*p* < 0.05), follow-up analyses were performed. In these, nine subsequent models were computed, examining the associations between BMI z-score and each of the ILR sets, including adjustment for the covariates. The co-efficient of the ILR_1_ was extracted from each of the nine models (and therefore each of the intensity bands) to examine the relationship of each intensity band (relative to all other intensity bands) with the BMI z-score [49]. Regression analyses were performed with the lmtest (version 0.9-40) [50] and car (version 3.0-12) [51] R packages, and model diagnostics were undertaken using the performance package (version 0.9.0) [52].

Where an intensity band (relative to the remaining bands) was significantly associated with BMI z-score (*p* < 0.05), this intensity band was then the focus of ‘one-for-remaining’ compositional isotemporal substitution analyses. These used predicted BMI z-scores for the initial ‘baseline’ average intensity band composition and compared them to the new BMI z-score predicted for subsequent compositions created using hypothetical time reallocations (1 to 20 min) between that intensity band and all the others. The estimated differences in BMI z-score for the reallocations of time to and from the selected average composition intensity band, and equally between the remaining intensity bands, were calculated by finding the difference between the two predicted BMI z-scores [53]. Adjustment for covariates was included in all the time reallocation analyses. Ninety-five percent confidence intervals (CIs) for the estimated differences in predicted BMI z-score were generated using the deltacomp R package (version 0.2.2, Mathsoft, Cambridge, MA, USA) [54]. The estimated difference in BMI z-score was considered significant when the 95% CI did not cover zero.

## 3. Results

### 3.1. Descriptive Results

Of the 1453 participants in the analytical sample, 57.5% were girls, 26.6% were overweight or obese, and two-thirds attended primary school (Table 1). Among boys, 71.3% lived in lower SES neighbourhoods (EIMD deciles 1–5) compared to 64.5% of girls. The participants were highly compliant in wearing the accelerometers, averaging 5.3 days of valid wear for 15.3 ± 1.0 h·day^−1^.

The geometric means of the intensity bands (linearly adjusted to collectively sum to 960 min) are presented in Table 2. Around 75% of waking hours were spent in the ‘inactive’ acceleration band of 0–50 mg. A linearly decreasing pattern of accumulated time in the intensity bands was observed, with fewer minutes spent in the higher intensity bands. The exceptions to this were the 350–700 mg and ≥700 mg bands, which were the sum of the combined 50 mg bands. The variation matrices of the time-use compositions representing the variability of the compositional dataset are included in Appendix A. 

### 3.2. Compositional Regression Analyses

BMI z-score was significantly associated with the intensity spectrum composition ILR_1_ coordinates (*F*(8,1430) = 11.9, *p* < 0.001) and sex (*F*(1,1430) = 10.6, *p* < 0.001) after the adjustment for covariates. Subsequent sex-stratified adjusted analyses revealed that BMI z-score was significantly associated with the intensity spectrum compositions for boys (*F*(8,1430) = 5.0, *p* < 0.001) and girls (*F*(8,1430) = 9.6, *p* < 0.001) (Appendix A). The unstandardised beta coefficients for each intensity band ILR_1_ determined which bands were the most dominant in the relationship with BMI z-score relative to time in all the remaining intensity bands. The ≥700 mg intensity band ILR1 was most strongly and inversely associated with BMI z-score (boys: *β*ILR_1_ = −0.77, *p* < 0.001; girls: *β*ILR_1_ = −0.71, *p* < 0.001) relative to the other ILRs in the composition (Table 3). Further, for girls, there were significant associations between BMI z-score and the 50–100 mg (*β*ILR_1_ = 1.39, *p* = 0.002) and 100–150 mg (*β*ILR_1_ = −2.55, *p* = 0.002) intensity band ILR_1_s relative to the remaining ones.

### 3.3. Compositional Isotemporal Substitution Analyses: One-to-Remaining Reallocations

The BMI z-scores for the baseline intensity band compositions (Table 2) were 0.72 units (boys) and 0.56 units (girls). Figure 1a shows the estimated differences in the boys’ BMI z-scores when 1 to 20 min were added to, and when 1 to 10 min (10 min was the maximum duration that could be reallocated from the ≥700 mg intensity band ILR_1_ because its geometric mean in the average composition was 11.1 min) were subtracted from—the baseline composition duration of the ≥700 mg intensity band (i.e., the most influential intensity band in Table 3) and redistributed equally among the remaining intensity bands (Appendix A). Reallocating time from the ≥700 mg intensity band was reflected by unfavourable estimated differences in BMI z-score that were greater than when the time was reallocated to this intensity band. For example, the estimated differences in BMI z-score when 10 min were reallocated to and from the ≥700 mg intensity band were −0.48 units (95% CI = −0.67, −0.29) and 1.71 units (95% CI = 1.03, 2.39), respectively.

Figure 1b–d present the estimated differences in the girls’ BMI z-scores following time reallocations involving the most influential intensity bands (50–100 mg, 100–150 mg, and ≥700 mg, respectively) and the remaining intensity bands. The reallocated time durations were 1 to 20 min, except for the ≥700 mg intensity band, where 1 to 5 min (5 min was the maximum duration that could be reallocated from the ≥700 mg intensity band ILR_1_ because its geometric mean in the average composition was 6.3 min) were subtracted from the baseline composition duration (Appendix A). Adding 20 min to the 50–100 mg band (Figure 1b) reflected an estimated difference of 0.26 BMI z-score units (95% CI = 0.10, 0.32) compared to –0.30 (95% CI = −0.49, −0.12) when 20 min were substituted from this band. The estimated differences in BMI z-score when 20 min were reallocated to and from the 100–150 mg intensity band (Figure 1c) were –0.81 (95% CI = −1.34, −0.28) and 1.18 (95% CI = 0.41, 1.96), respectively. Reallocating 5 min to the ≥700 mg intensity band yielded a predicted difference in BMI z-score of −0.39 units (95% CI = −0.51, −0.28) compared to 1.06 (95% CI = 0.76, 1.36) when 5 min were substituted from the ≥700 mg band and reallocated equally among the remaining bands (Figure 1d).

## 4. Discussion

In this study we examined the associations between the physical behaviour intensity spectrum composition and BMI z-score across a wide age range of English youth. This is the first pooled individual participant data analysis in which harmonised raw acceleration data derived from different accelerometer models have been analysed in this way. For boys and girls, the highest intensity band (≥700 mg) was the most dominant in the relationship with BMI z-score (relative to the time spent in the remaining intensity bands). Respectively, time spent by girls in the 50–100 mg and 100–150 mg intensity bands was also positively and inversely associated with BMI z-score. Reallocating time between these dominant intensity bands and equally among the remaining bands resulted in predicted increases in BMI z-score when minutes were taken from the 100–150 mg and ≥700 mg intensity bands and decreases in BMI z-score when minutes were substituted from the 50–100 mg band. The predicted BMI z-score increases were larger than the predicted decreases when corresponding time reallocations were made to the 100–150 mg and ≥700 mg intensity bands. 

Our main finding that the strongest associations with BMI z-score were from the highest intensity band concurs to some extent with the previous compositional analysis of the intensity spectrum in school-aged children, although it should be noted that this study did not focus on BMI z-score as the health outcome. Aadland et al.’s analysis in Norwegian 10-year-olds used hip-worn ActiGraph counts data and reported that the time in the 7500–7999 counts·min^−1^ intensity band was significantly and negatively associated with cardiometabolic risk [11]. The lack of direct comparability between proprietary ActiGraph hip counts data and ActiGraph, GENEActiv, and Axivity raw acceleration wrist data makes it difficult to translate Aadland et al.’s findings to ours. Although not specific to BMI z-score or other adiposity outcomes, these authors have also examined the accelerometer data intensity spectrum using multivariate pattern analysis [9]. They found that time spent in the 5000–7000 counts·min^−1^ intensity bands (ActiGraph hip counts data) was most strongly associated with children’s metabolic health [55,56]. Similar conclusions about the higher intensity bands (4000–5000 counts·min^−1^) were made in a more recent pooled analysis of data from over 11,000 children [12]. Although we also applied an intensity spectrum approach, it is difficult to make direct comparisons with the findings from these multivariate pattern analysis studies because of the different health outcomes studied and the aforementioned methodological and analytical differences between them. However, notwithstanding these differences, the common conclusion is that, when using the full range of available acceleration data to examine the relationships with indicators of adiposity or metabolic risk in youth (of which adiposity is a contributory factor), the magnitude of associations is stronger for time spent in the highest, rather than lower, intensity ranges.

In contrast to our analysis of the physical behaviour intensity spectrum, almost all previous compositional analyses of youth physical behaviours have used three or four component compositions defined by published cut-points. These have consistently reported MVPA to be most strongly and negatively associated with adiposity indicators relative to other physical behaviours [20,21,24,25,27,53]. However, when using MVPA as the highest intensity component in physical behaviour compositions, important parts of the accelerometer data are lost between the lower MPA threshold and the upper levels of VPA. This might lead to a loss of information that may be a hallmark of childhood obesity, making it unclear which specific intensities are most strongly related to adiposity. Some light has been shed on this question by compositional analyses of waking hours sedentary time, LPA, MPA, and VPA among Czech [57,58] and American youth [59], which observed VPA to be negatively and significantly associated with adiposity indicators relative to the other behaviours. These results support ours and demonstrate how the commonly reported finding that MVPA has the strongest influence on adiposity-related indicators relative to other physical behaviours actually masks the potentially more important contribution of higher intensity activities, while simultaneously amplifying the influence of lower intensity activities that fall within the MVPA range, as determined by the chosen absolute intensity cut-points. The use of absolute intensity cut-points derived from specific calibration study samples and protocols increases the likelihood of misclassifying time estimates in each physical behaviour [60], which can result in the over- and under-estimation of intensity-specific activity patterns [61]. By using multiple smaller incremental intensity bands rather than three-to-four broad cut-point categories, we allowed for a more nuanced description and analysis of participants’ activity patterns. Moreover, our analysis demonstrates the utility of compositional analysis for raw accelerometer data presented as an intensity spectrum, which further supports the integration of multiple physical behaviours and intensities to promote health and wellbeing among youth [59]. 

Among girls, BMI z-score was positively associated with time in the 50–100 mg intensity band, which may reflect a combination of low energy expenditure seated postures with arm movements and/or standing stationary postures [62]. In contrast, time spent in the hypothesised ‘sedentary’ 0–50 mg intensity band [62], relative to the other bands, was not significantly associated with BMI z-score. We used a standardised waking hours ‘day’ (i.e., 07.00 to 23.00); thus, it was not possible to differentiate between sedentary time and sleep within the 0–50 mg intensity band. As a consequence, some participants may have been asleep after 07:00 or before 23:00, and, as sleep is favourably associated with obesity risk [63], this may have confounded the strength of any positive associations between BMI z-score and sedentary time captured in the 0–50 mg band. It may also partly explain why accumulated very low intensity time in the 50–100 mg band had a significant and stronger positive association with girls’ BMI z-scores. Significant negative associations were observed between girls’ BMI z-scores and time in the 100–150 mg intensity band. Based on the available published raw acceleration data from wrist-worn devices, this lower intensity band may reflect slow-medium paced walking, akin to LPA [10]. Compositional analysis studies using count cut-points have reported both positive associations [24,27] and negative associations between LPA and BMI z-scores [57]. Moreover, an earlier non-compositional study observed negative relationships between LPA and children’s fat mass [64]. In most free-living studies, LPA represents the longest accumulated duration of waking hours physical activity and typically encompasses ActiGraph counts in the 100 to 2000+ counts·min^−1^ range. This likely includes some misclassification of sedentary time and MPA, which, in addition to differences in analytical approaches (i.e., non-compositional vs. compositional), provides insight into why the findings from previous cut-points studies lack agreement. Applying compositional analysis, which accounts for the mutually exclusive relationships between activity intensities, with higher-resolution intensity bands rather than a limited number of cut-points can advance this area of research by allowing for a more detailed examination of activity intensity associations with health outcomes.

The significant associations between lower intensity bands and BMI z-scores were only apparent in girls, who are typically less active and less fit than boys [21]. For our pooled sample, the 100–150 mg intensity band, in which girls accrued more time than boys, may have represented a higher relative intensity for some girls that was favourably associated with BMI z-score. This is consistent with previous research showing volume of physical activity to be more strongly associated with health outcomes in lower active and less fit groups, while among higher active and more fit groups, associations were strongest with increasing physical activity intensity [65]. Furthermore, the associations may have been influenced by the environmental contexts in which the physical behaviours occurred. For example, significant negative associations were recently reported between out-of-school LPA and adiposity in girls but not boys, while associations with school time LPA were positive and non-significant irrespective of sex [66]. 

We found that when hypothetical time reallocations were modelled between one intensity band and equally among the remaining bands, the most favourable predicted differences in boys’ and girls’ BMI z-scores were when time was added to the ≥700 mg intensity bands. Consistent with previous compositional analyses, the predicted changes in BMI z-score were asymmetrical [20,21,24,25,27,53]. This demonstrates that the potential detrimental health effects of taking time away from activity above the 700 mg intensity band and redistributing it equally to the remaining bands were greater than the beneficial effects of adding time at this higher intensity. The relative contributions of each intensity band to the waking hours day provide some insight into these asymmetrical relationships. Taking time away from the ≥700 mg intensity band, which contributed to 1.2% (boys) and 0.7% (girls) of the 16 h waking day, is a substantially larger relative change than taking time from the 50–100 mg intensity band, for example, which contributed to 10.7% (boys) and 11.2% (girls) of the day [21]. The observed predicted differences in BMI z-score were also not linearly related to the durations of reallocated time in the ≥700 mg and 100–150 mg bands (girls only). This has been consistently observed in previous compositional analysis studies [21,25,53] and reflects findings from experimental research whereby diminishing health benefits are predicted by marginal increases in physical activity [67]. Moreover, the wider 95% confidence intervals corresponding to the girls’ 100–150 mg time reallocations indicate greater variability and relatively less precision in the predicted BMI z-score differences compared to those from the ≥700 mg band reallocations. This complements the consistent relationship between the higher intensity activity and health outcomes reported in compositional analysis studies [20,21,24,25,27,53,57,59] and underscores the importance of promoting, providing, and not withholding developmentally appropriate and enjoyable opportunities for youth to be active at these intensities, even for short accumulated durations (e.g., through active play, sports, physical education, HIIT, etc.). In our sample, as little as 3 min (girls) and 5 min (boys) of additional time spent in the ≥700 mg intensity band relative to the remaining bands were associated with predicted decreases in BMI z-score of 0.26 and 0.28 units, respectively. These values are greater than the BMI z-score mean differences between the intervention and control groups for adolescents reported in a recent Cochrane review of obesity prevention interventions [5] and those observed in an effective school-based healthy weight intervention for children [68]. Hence, accumulating time spent being active at these higher intensities is advocated as being beneficial for health in youth. 

This study employed a large pooled individual participant dataset comprising primary and secondary school youth. Assessments of the physical behaviours from different devices were used to generate raw acceleration data that enabled the analysis of the physical behaviour intensity spectrum rather than a small number of activity intensities derived from absolute intensity cut-points. Compositional data analysis with ‘one-for-remaining’ isotemporal substitutions enabled the associations with BMI z-score and predicted changes to be presented. There were, however, limitations that should be considered when interpreting the results. The contributing studies were cross-sectional, so causality between the intensity spectrum compositions and BMI z-score associations cannot be inferred, and the possibility of bi-directional associations is acknowledged. Further, the use of a cross-sectional analysis meant that the estimated differences in BMI z-score reflected more of a sample shift in intensity spectrum time allocations than actual differences for individual participants [69]. As the resolution of intensity spectrum bands can result in different interpretations of the relationship between activity intensity and health [14], our choice of nine intensity spectrum bands may have influenced the associations with BMI z-score. In future, using a higher resolution intensity spectrum with emerging analytical techniques such as functional data analysis may help improve our understanding of the relationships between specific acceleration ranges and health outcomes [9]. Although the analyses were adjusted for sociodemographic and methodological variables, it is possible that residual confounding from other non-measured variables such as sleep may have influenced the results. Lastly, the regional sample limits the generalisation of the findings to the wider UK and beyond.

## 5. Conclusions

This innovative compositional data analysis of the physical activity intensity spectrum in 5-to-16-year-olds is the first to use pooled raw acceleration data from different devices. Time in the highest intensity band (≥700 mg), relative to the remaining intensity bands, was significantly and negatively associated with BMI z-score, irrespective of sex. ‘One-for-remaining’ time reallocations involving the ≥700 mg intensity band indicated that the asymmetrical estimated differences in BMI z-score were meaningful at even modest volumes of reallocated time. The consistency of the results with previous findings suggests that pooling raw acceleration data from different devices was appropriate, which highlights the utility of this approach. Our novel results highlight the utility of the full physical activity intensity spectrum over a priori-determined absolute intensity cut-point approaches and further emphasise the benefits of promoting higher intensity physical activity for health in youth. Furthermore, they can provide researchers, public health professionals, and physical activity deliverers with important insights to inform obesity prevention intervention design and physical activity programming.

## Figures and Tables

**Figure 1 ijerph-19-08778-f001:**
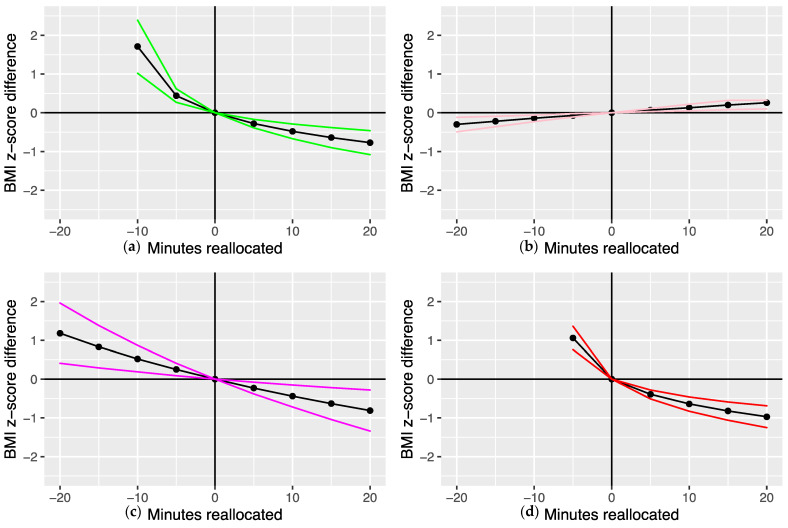
(**a**–**d**) Predicted differences in BMI z-score for the time reallocations between the most dominant intensity bands and the remaining intensity bands. Note: (**a**) ≥700 mg intensity band (boys); (**b**) 50–100 mg (girls); (**c**) 100–150 mg (girls); (**d**) ≥700 mg (girls). Note: coloured lines represent the lower and upper boundaries of the 95% confidence intervals.

**Table 1 ijerph-19-08778-t001:** Participants’ descriptive characteristics (M (SD) or %).

	All *(n=* 1453)	Boys (*n* = 624)	Girls (*n* = 829)
Age (years)	10.5 (2.6)	10.0 (2.6)	10.8 (2.5)
Height (cm)	142.1 (16.2)	139.8 (16.8)	143.7 (15.6)
Weight (kg)	39.6 (14.8)	37.3 (14.4)	41.3 (15.0)
BMI (kg·m^−2^)	19.0 (3.9)	18.4 (3.6)	19.4 (4.1)
BMI z-score	0.51 (1.24)	0.53 (1.29)	0.49 (1.21)
Weight status			
Normal weight (%)	73.4	74.2	72.9
Overweight/obese (%)	26.6	25.8	27.1
EIMD decile			
Deciles 1–5 (%)	67.4	71.3	64.5
Decile 6–10 (%)	32.6	28.7	35.5
School type			
Primary (%)	66.3	72.4	61.8
Secondary (%)	33.7	27.6	38.2

Note: BMI = body mass index; EIMD = English Indices of Multiple Deprivation.

**Table 2 ijerph-19-08778-t002:** Geometric means of the time spent in the intensity spectrum bands.

	All (*n* = 1453)		Boys (*n* = 641)		Girls (*n* = 862)	
Intensity Band	min·day^−1^	%	min·day^−1^	%	min·day^−1^	%
0–50 mg	722.9	75.3	719.9	74.9	724	75.5
50–100 mg	105.7	11	102.6	10.7	108	11.2
100–150 mg	51.1	5.3	49.6	5.2	52	5.4
150–200 mg	27.8	2.9	27.8	2.9	28	2.9
200–250 mg	15.4	1.6	16.1	1.7	15	1.5
250–300 mg	9.1	0.9	9.8	1	9	0.9
300–350 mg	5.8	0.6	6.4	0.7	5	0.6
350–700 mg	14.3	1.5	16.7	1.7	13	1.3
≥700 mg	8.0	0.8	11.1	1.1	6	0.6

Note: mg = milligravitational units.

**Table 3 ijerph-19-08778-t003:** Regression model results for boys and girls in assessing the compositional association between each intensity spectrum band ILR_1_ and BMI z-score, relative to the remaining intensity bands, with adjustment for SES, centred age, accelerometer model, and accelerometer sampling frequency.

	Boys			Girls		
Intensity Band ILR_1_ (mg)	*β*ILR_1_	95% CI	*p*	*β*ILR_1_	95% CI	*p*
0–50 mg vs. remaining	−0.20	−0.61, 0.21	0.34	0.03	−0.28, 0.34	0.87
50–100 mg vs. remaining	−0.74	−1.92, 0.44	0.22	1.39	0.53, 2.25	0.002
100–150 mg vs. remaining	−0.34	−2.65, 1.97	0.76	−2.55	−4.18, −0.92	0.002
150–200 mg vs. remaining	0.88	−1.96, 3.72	0.55	1.87	−0.05, 3.79	0.06
200–250 mg vs. remaining	−0.50	−3.24, 2.24	0.72	−1.64	−3.56, 0.28	0.11
250–300 mg vs. remaining	−1.75	−4.14, 0.64	0.15	1.04	−0.51, 2.59	0.19
300–350 mg vs. remaining	1.39	−0.61, 3.39	0.17	−0.09	−1.38, 1.20	0.89
350–700 vs. remaining	0.57	−0.41, 1.55	0.26	0.39	−0.39, 0.99	0.26
≥700 mg vs. remaining	−0.77	−1.08, −0.46	<0.001	−0.71	−0.91, −0.51	<0.001

Note: BMI = body mass index; CI = confidence interval; ILR = isometric log-ratio; mg = milligravitational units.

## Data Availability

The dataset analysed during the current study is available from https://osf.io/kae5r/. The R code used for the data analysis is available from https://osf.io/ze4nk/.

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
