# Peer review of "The Physical Behaviour Intensity Spectrum and Body Mass Index in School-Aged Youth: A Compositional Analysis of Pooled Individual Participant Data"

_ijerph, 2022, doi:10.3390/ijerph19148778_

Round 1

Reviewer 1 Report

Dear authors, thank you for your valuable and meaningful work. In this study, nine acceleration intensity bands (ranging from 0-50 mg to ≥700 mg) were allocated to generate time-use combinations by summarizing the original data based on different accelerometer studies. The relationship between the components of intensity band and Z score of body mass index was evaluated by multiple regression. It is the estimated difference of time substitution of Z value of body mass index after time redistribution between component intensity bands. There was a strong negative correlation between ILR of intensity band ≥700 mg and Z score of body mass index (p<0.001). Read the full text carefully, and there are several aspects that need further improvement.

1. An important conclusion is that "700 mg intensity band ILR has a strong negative correlation with body mass index Z score (p<0.001)". But this conclusion is still difficult to support the importance of this research. This conclusion has been confirmed in many related studies on physical activity and body mass index of school-age youth.

2. In Figure 1. A-D. Is it 1b: 50-100 mg (girls) or 1b: 50-100 mg (boys)?

3. The research results emphasize the effectiveness of the whole physical activity intensity spectrum relative to the absolute intensity cut-off point method determined a priori-however, the research results are not enough to support this conclusion.

4. If the author wants to emphasize the spectrum of physical activity intensity through the method of absolute intensity cut-off point determined a priori, it is necessary to describe more clearly the current reported research and more detailed evidence on the relationship between the cut-off point of physical activity intensity and the body mass index of school-age youth in the preface. On the other hand, it is necessary to add systematic evaluation to the research methods, and the research results can be added to tables for detailed display.

Reviewer 2 Report

Authors study the very important topic of children's physical activity levels and BMI. Future studies may be able to use this information and determine if their is an association between physical activity, BMI, and metabolic health.

41: When even modest durations of time in this band were reallocated, (add comma) 

50-51: Overweight and obesity in children and adolescents (from here referred to as youth), significant risk factors for health and wellbeing, continue to increase in prevalence.

Do authors mean to state that overweight and obesity continue to increase in prevalence? or health and well-being continue to increase in prevalence?

68: of it is used (to what does it refer?)

97: Focusing on 5-to-16-year olds, (add comma) our novel study

110: omit For

153-4: Participants’ accelerometer data were excluded from analyses if post-calibration er- 153 ror was >10 milligravitational units (mg) and/or <3 days of valid wear (i.e., ≥600 min·day– 154 1) were available.

So does that mean if someone had valid wear for at least 5 days, the data was included? Line 167 states "Time in the nine bands summed to 960 minutes."

How is that possible if participants with only 5 or 6 days of activity were included? Perhaps you want to explain the linear adjustment (Table 2) in this section of paper.

188: z-score was

215:  the remaining intensity bands, (add comma) were

223-224: change to had overweight or obesity. They are diseases

236: Table 1: Big difference between BMIs in the lower range of the overweight category to BMIs in higher range of obesity category. Was there any significance when overweight and obesity were considered separately?

252: relative (to) the remaining ones 

284: Figure 1 - label the boxes (which is 1b and which is 1c? - are you going down or horizontally?)

297: , relative the remaining ones. Omit comma

318: However, notwithstanding these differences, (add comma)

335-336: reported finding that MVPA has the strongest influence on adiposity-re- lated indicators, (add comma) relative to other physical behaviours, 

344: analysis

366-7: In most free-living studies, (add comma) LPA represents the longest 

370:  (i.e., non-compositional vs. compositional), (add comma) provides 

371: Applying compositional analysis which account(s) for

373-4:  of cut-points, (add comma) can

376: omit these

377: girls, who

Add comma if you are trying to convey that all girls tend to be less physically active than boys. Currently this reads that the significant associations are among the girls in this study who are less active/fit.

346-347, 375, 380-383, 404, 408, 454, etc...   Focus of current paper is intensity of physical activity and BMI; there is no study of an association between physical activity and metabolic health/health outcome. Yet authors mention the association of physical activity and metabolic health several times throughout the paper. The extrapolation of PA to BMI and subsequently, health outcomes, can be mentioned in the discussion as a future study but should not be mentioned as correlating in the current study. Stick to PA and BMI

396: provides--> provide

399: which contributed (to) 10.7%

436: In (the) future, using

Alternate between BMI z-score and z-scores throughout manuscript - be consistent

Reviewer 3 Report

In this study the  authors examined associations between the physical behaviour intensity  spectrum composition and BMI z-score across a wide age range of English youth, for the first time.  The title and abstract cover the main aspect of the work.  The introduction provide background and information relevant to the study. However, the abstract could be improved (has no background, should add a proposition explaining the importance of the research theme to increase the reader interest). The methods are not very clear and should be improved. For example, it is not clear how or where were the included studies found. Please specify more clearly the methods of inclusion. The research theme is novel in general, however, the practical applicability of these results it is not clearly stated. Please enhance the conclusion by explaining why the practitioners should use these results.

Also, I have several observations to make below.

The spelling of behaviour/behaviours is a non-American variant. For consistency, consider replacing it with the American English spelling. 

Line 49-51: This sentence appears to be missing a subject. Consider adding a subject or rewriting the sentence.

The spelling of unfavourable is a non-American variant. For consistency, consider replacing it with the American English spelling.

The spelling of emphasised is a non-American variant. For consistency, consider replacing it with the American English spelling. 

The expression "TV viewing in particular" should be rewritten, explaining the word TV.

Reviewer 4 Report

The article deals with a very important aspect of physical activity of children and children in the long-term assessment. The use of measuring sensors for the assessment of the continuity of physical activity is very interesting, but the authors presented their work in a very incomprehensible way. the content is very technical without introduction, there is no explanation of particular terms which are not commonly used in assessing physical activity such as spectrum, spectrum which is more associated with assessment used in physics to assess waves or light.

The abstract is very obscure and does not encourage further reading, and it certainly does not indicate that the authors have focused on assessing the intensity of physical breakdown in continuous monitoring.

There is no information on the selection of the study group, whether they are randomly selected schools or those in which the authors work.

Why children from primary and secondary school were included in one group - they have a different burden in terms of learning and thus the time spent in a sitting position.

No information on the temporal relation of activity intensity

Lack of information as to which values ​​are desired - since this is an innovative method, it is worth presenting the scale on which young people should maintain themselves in order to stay healthy. It is not known whether the results are good or bad

the work in its present form is not suitable for printing

is targeted at a very small group of people who use this method of assessing the intensity of physical activity

Round 2

Reviewer 1 Report

The author has responded to my opinions, which basically meet the requirements. Although some places hold different opinions, on the whole, they have met the requirements of publication. Although it is feasible to extensively absorb other people's raw data, the most ideal scheme is to participate in the experiment yourself, so as to be more convincing.